# Dopamine, Norepinephrine and Serotonin Participate Differently in Methylphenidate Action in Concomitant Behavioral and Ventral Tegmental Area, Locus Coeruleus and Dorsal Raphe Neuronal Study in Young Rats

**DOI:** 10.3390/ijms242316628

**Published:** 2023-11-22

**Authors:** Cruz Reyes-Vasquez, Zachary Jones, Bin Tang, Nachum Dafny

**Affiliations:** 1Physiology Department, Medical School, National Autonomous University of Mexico, Mexico City 04510, Mexico; 2Department of Neurobiology and Anatomy, McGovern Medical School, The University of Texas Health Science Center, 6431 Fannin Street, Houston, TX 77030, USA

**Keywords:** monoamines, brain motive circuit, psychostimulant

## Abstract

Methylphenidate (MPD), known as Ritalin, is a psychostimulant used to treat children, adults, and the elderly. MPD exerts its effects through increasing concentrations of dopamine (DA), norepinephrine (NE), and serotonin (5-HT) in the synaptic cleft. Concomitant behavioral and neuronal recording from the ventral tegmental area (VTA), locus coeruleus (LC), and from the dorsal raphe (DR) nucleus, which are the sources of DA, NE, and 5-HT to the mesocorticolimbic circuit, were investigated following acute and repetitive (chronic) saline, 0.6, 2.5, or 10.0 mg/kg MPD. Animals received daily saline or MPD administration on experimental days 1 to 6 (ED1–6), followed by a 3-day washout period and MPD rechallenge on ED10. Each chronic MPD dose elicits behavioral sensitization in some animals while inducing behavioral tolerance in others. The uniqueness of this study is in the evaluation of neuronal activity based on the behavioral response to chronic MPD. Neuronal excitation was observed mainly in brain areas of animals exhibiting behavioral sensitization, while neuronal attenuation following chronic MPD was observed in animals expressing behavioral tolerance. Different ratios of excitatory/inhibitory neuronal responses were obtained from the VTA, LC, or DR following chronic MPD. Thus, each brain area responds differently to each MPD dose used, suggesting that DA, NE, and 5-HT in the VTA, LC, and DR exert different effects.

## 1. Introduction

Methylphenidate (MPD) was synthesized in 1944 and was patented in 1954. Its use started in 1955 as a medication for the elderly to treat locomotor fatigue activities, aiming to prevent falling, sleep disorders, and memory recognition deficits, and is now a widely used drug by all ages. Knowledge about this drug is still quite vague as it remains unclear how it specifically affects neuronal and behavioral activities [1,2,3,4,5,6] MPD is a central nervous system (CNS) stimulant known for its effectiveness in treating behavioral disorders, such as attention deficit hyperactivity disorder (ADHD), and improving cognitive and social functions. More recently, its unregulated use by ordinary young and adults for cognitive enhancement, concentration improvement, and increased productivity and academic performance has become more popular, being coined as a “study drug” or a “smart drug’ [7,8,9,10], as well as its abuse for recreational purposes [11,12]. The increasing usage of MPD in both young and adult populations has sparked an interest in investigating its effects in younger subjects, as their brains are still in developmental stages, including synaptic changes and pruning [13]. It is therefore important to study the properties of MPD in young animal models, when their brains are still going through developmental changes, to verify its effect. This has become a serious public health concern as the neurological and psychiatric consequences of unrestricted usage of psychostimulants are not known [14,15,16]. Previous investigations by our group have studied the properties of the MPD dose response on the VTA [17], LC [18], and DR [19] in adult animals. Knowledge about this drug is still quite vague as it remains unclear how it specifically affects neuronal and behavioral activities [1,2,6,20]. It was reported that the mechanism of MPD on the CNS is based on its effects on DA and NE transmission in the brain, and limited studies reported that 5-HT also plays an important role in the overall effects of MPD and its ability to enhance cognitive function [19] The aim of this study is to investigate the role of MPD on the three brain areas that are the main sources of DA, NE, and 5-HT with regard to the mesocorticolimbic system, and to compare their roles using simultaneous neuronal and behavioral assays in freely behaving animals. This will help to gain a deeper understating into the function and response roles of these different brain regions to different doses of MPD in an acute and chronic setting.

MPD exerts its cognitive effects by binding to the dopamine (DA), norepinephrine (NE), and serotonergic (5-HT) transporters (DAT, NET, and SERT) in the synaptic cleft, preventing their reuptake back into the presynaptic terminal, increasing the concentrations within the synaptic cleft [21,22,23,24,25]. This effect occurs in regions of the brain that are involved in the motivation and reward circuit: prefrontal cortex (PFC), caudate nucleus (CN), ventral tegmental area (VTA), nucleus accumbens (NAc), locus coeruleus (LC), and dorsal raphe (DR). [23,24,26,27,28]. While the use of MPD in the treatment of ADHD has been clinically established, its use for cognitive enhancement and recreation beyond its intended application has become more prevalent for many reasons. Studies have shown that over 20% of college students on USA campuses use MPD and other stimulants to improve cognitive function, thus improving academic performance, as well as for recreation for its euphoric properties [11]. This is of great concern as MPD abuse can have potentially lethal outcomes, especially if taken in the intravenous or intranasal route, which is why it is essential to study the effects of MPD on otherwise healthy subjects in detail [9,11,29,30]. This has become a serious public health concern as the neurological and psychiatric consequences of unrestricted usage of psychostimulants are not known [14,15,16].

By recording data from each brain structure separately, it appeared that MPD exerts similar effects in all of the above brain structures. It is important to verify this conclusion and perform this study of neuronal recordings from the above three brain areas simultaneously, concomitantly with its effects on behavioral activity following acute and chronic consumption to further elucidate the accuracy of the previous assumptions. Indeed, this study demonstrated that our previous conclusion was wrong. Repetitive (chronic) exposure to psychostimulants, such as MPD, has been shown to elicit dose-dependent behavioral sensitization or tolerance in animal models [13]. Sensitization refers to the phenomenon in which repeated administration of the same dose of drug elicits a greater effect as compared to its initial (acute) effects, while tolerance refers to the phenomenon in which an increasing amount of drug is required to elicit the same response as compared to the acute effects. These two behavioral expressions occur as a result of chronic drug exposure. Sensitization and tolerance represent key experimental biomarkers that indicate whether a drug has a potential for abuse [13,31] and may be critical in understanding the long-term effects of MPD in freely behaving animals. The aim of this study is to investigate the role of MPD in the three brain areas that are the main sources of DA, NE, and 5-HT with regard to the mesocorticolimbic system using simultaneous neuronal and behavioral assays in freely behaving animals. This will help to further elucidate the roles that these different brain regions have concerning their functions and responses to different doses of MPD in an acute and chronic setting.

## 2. Results

### 2.1. Behavioral Response to Acute and Chronic MPD Exposure

One hundred and fifty-three young rats with a post-natal age of 30 (P 30) were purchased from Harlem Ind., USA, and were divided into four groups of 8, 45, 49, and 51 rats, treated with saline and 0.6, 2.5, or 10.0 mg/kg MPD, respectively.

### 2.2. Control

Eight rats were used as a control to test for the effects of injection volume and handling procedure. These animals received saline treatment on ED1–ED6 and ED10 (Table 1) to test the effects of acute and repetitive (chronic) saline administration. All animals exhibited similar locomotor activity behaviors before and after saline injection with no significant changes in behavior as measured by the number of movements (NM), total distance (TD) traveled in cm, and number of stereotypic movements (NOS) (Figure 1). This observation indicates that injection and handling does not affect locomotor behavioral activities and that any significant differences in measured locomotor activity after MPD exposure compared to the saline (baseline) activity can be attributed to the effects of the tested drug.

### 2.3. Effect of Acute and Chronic 0.6 mg/kg MPD (Figure 2, Top Panel, N = 45 All Group)

Forty-five young rats starting at age P-40 received acute and chronic administration of 0.6 mg/kg MPD. There was a significant (*p* < 0.05) increase in NM after acute MPD exposure on experimental day 1 (ED1) compared to ED1 baseline (BL); (ED1 MPD/ED1 BL). There was a significant (*p* < 0.05) decrease in NM in all animal groups when comparing BL activity on ED10 to ED1 BL (ED10 BL/ED1 BL) after six daily 0.6 mg/kg MPD injections and three washout days, and there was a significant (*p* < 0.05) increase in NM when comparing the chronic effect of 0.6 mg/kg to acute MPD exposure (ED10 MPD/ED1 MPD) (Figure 2, top left panel). The animals were then sorted into subgroups based on their individual behavioral response to chronic MPD as compared to acute MPD (ED10 MPD/ED1 MPD) in relation to sensitization (N = 23 sensitized group) and tolerance (N = 22 tolerance group).

**Figure 2 ijms-24-16628-f002:**
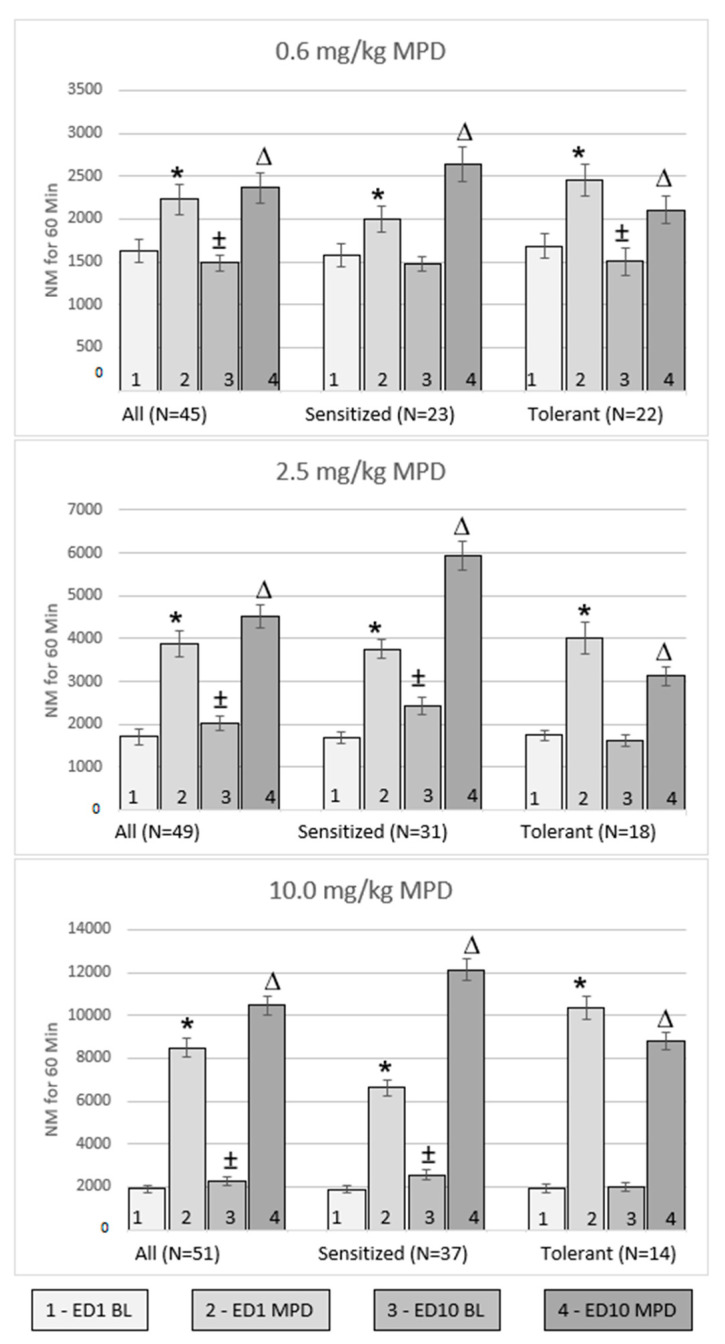
Summary of the number of movements (NM) in relation to initial saline injections (ED1 BL) (number 1), acute MPD (ED1 MPD) (number 2), ED10 BL activity after saline injections (number 3), and the activity following chronic MPD exposure (ED10 MPD) (number 4) at each respective MPD dose. For each dose, the left histogram represents the total activity of all animals, the middle histogram represents the activity of animals that expressed sensitization, and the right histogram represents the activity of animals that expressed behavioral tolerance. N = no. of animals in each group; ED1 BL = activity following initial saline injection on ED1; ED1 MPD = activity following acute MPD exposure on ED1; ED10 BL = activity following saline injection on ED10 after MPD administration during ED2–6; ED10 MPD = activity following chronic MPD exposure on ED10. * Represents a significant (*p* < 0.05) difference when comparing ED1 MPD to ED1 BL; ± represents significant (*p* < 0.05) differences comparing ED10 BL/ED1 BL; ^Δ^ represents a significant (*p* < 0.05) difference comparing ED10 MPD/ED1 MPD. The figure shows the acute effect of MPD, the effect of repetitive (chronic) MPD on ED10 baseline (BL) activity after chronic MPD exposure, and 3 washout days, as well the effect of chronic MPD on ED10.

In animals that exhibited behavioral sensitization (i.e., increased NM after chronic MPD exposure as compared to the initial MPD dose (ED10 MPD/ED1 MPD)), there was a significant (*p* < 0.05) increase in NM after acute 0.6 mg/kg MPD exposure, but no significant change in ED10 BL NM when compared to ED1 BL (ED10 BL/ED1 BL). There was a significant (*p* < 0.05) increase in NM after chronic 0.6 mg/kg MPD exposure as compared to acute MPD (ED10 MPD/ED1 MPD), i.e., these animals expressed behavioral sensitization (Figure 2, top center panel).

In animals that exhibited behavioral tolerance (i.e., decreased NM after chronic MPD exposures as compared to initial MPD effect), there was a significant (*p* < 0.05) increase in NM after acute MPD (ED1 MPD/ED1 BL) and a significant (*p* < 0.05) decrease on ED10 BL/ED1 BL. There was a significant (*p* < 0.05) decrease in NM after chronic 0.6 mg/kg MPD as compared to acute MPD (ED10 MPD/ED1 MPD); however, there was still a significant (*p* < 0.05) increase compared to ED1 MPD exposure (Figure 2, top right panel), indicating tolerance expression.

### 2.4. Effect of Acute and Chronic 2.5 mg/kg MPD (Figure 2, Middle Left Panel, N = 49 All Group)

Forty-nine animals received acute and chronic administration of 2.5 mg/kg MPD. There was a significant (*p* < 0.05) increase in NM after acute and chronic 2.5 mg/kg MPD exposure on ED1 MPD/ED1 BL, ED10 BL/ED1 BL, and ED10 MPD/ED1 MPD, respectively (Figure 2, all group).

Animals that exhibited behavioral sensitization (N = 31) (Figure 2, middle center panel) responded similarly to acute and chronic MPD treatment as the all animals group did before sorting. There was a significant (*p* < 0.05) increase in NM after both acute and chronic 2.5 mg/kg exposure, and a significant (*p* < 0.05) increase in NM on ED10 BL/ED1 BL (Figure 2, middle center panel).

Animals exhibiting behavioral tolerance (N = 18) (Figure 2, middle right panel) exhibited a significant (*p* < 0.05) increase in NM after acute 2.5 mg/kg MPD exposure, no change on ED10 BL/ED1 BL (Figure 2, middle right panel), and a significant (*p* < 0.05) decrease comparing ED10 MPD/ED1 MPD, and still showed a significant (*p* < 0.05) increase as compared to the initial MPD effects, i.e., expressing tolerance.

### 2.5. Effect of Acute and Chronic 10.0 mg/kg MPD (Figure 2, Bottom Panel, N = 51 All Group)

Fifty-one animals received acute and chronic administration of 10.0 mg/kg MPD. There was a significant (*p* < 0.05) increase in NM after both acute and chronic 10.0 mg/kg MPD exposure (ED1 MPD/ED1 BL and ED10 MPD/ED1 MPD and in ED10 BL/ED1 BL in the all animals group (Figure 2, bottom left panel).

After sorting the animals following chronic MPD exposure compared to the initial MPD effect (ED10 MPD/ED1 MPD), 37 animals exhibited behavioral sensitization and 14 animals expressed behavioral tolerance. Both groups (behavioral sensitization and behavioral tolerance) showed the same response pattern as the all animals group to acute MPD exposure (ED1 MPD/ED1 BL). For the animals from the sensitized group, the ED10 BL after six daily MPD exposures and three washout days (ED10 BL/ED1 BL) exhibited a significant (*p* < 0.05) increase in their NM (Figure 2, bottom middle and right panel); in the animals from the tolerance group, chronic MPD (ED10 BL/ED1 BL) did not change the ED10 BL and showed a significant (*p* < 0.05) decrease compared to ED1 MPD and a significant (*p* < 0.05) increase compared to the initial MPD effect, i.e., expressing tolerance. Similar observations with minor non-significant fluctuations were observed for TD and NOS behavioral locomotor activities.

### 2.6. Neuronal Responses to Acute and Chronic MPD Exposure (Table 2)

Table 2 summarizes the total significant (*p* < 0.05) responsiveness of the VTA, LC, and DR neurons to saline and 0.6, 2.5, and 10.0 mg/kg MPD. A total of 1460 histologically confirmed neurons (480 VTA, 476 LC, and 504 DR, respectively) were recorded (Table 2). A total of 45, 56, and 57 from the VTA, LC, and DR following saline and 435, 420, and 447 neuronal units were evaluated following saline and 0.6, 2.5, and 10.0 mg/kg MPD, respectively.

**Table 2 ijms-24-16628-t002:** Summary of the number of the neuronal recordings from all animal groups from the ventral tegmental area (VTA), locus coeruleus (LC), and the dorsal raphe (DR), as well as their total number of responsive neurons and their percentage (%) in relation to the initial (acute) methylphenidate (MPD). Total significant (*p* < 0.05) responsiveness.

	VTA	LC	DR
MPD Dose (mg/kg)	N Resp	N Resp	N Resp
Saline	45-2 (4.4%)	56-4 (7.1%)	57-3 (5.3%)
0.6	141-40 (28.3%)	134-83 (61.9%)	137-71 (57.8%)
2.5	142-78 (54.9%)	146-92 (63.0%)	142-89 (62.7%)
10.0	152-105 (69.1%)	140-128 (914%)	168-134 (79.8%)

### 2.7. Effect of Saline on VTA, LC, and DR Neurons Recorded from All Group

A total of 45 VTA, 56 LC, and 57 DR neurons were recorded following saline treatment. The firing rates show little to no significant difference after saline administration within and between the three brain regions on ED1 and ED10, suggesting that saline exposure is an appropriate control. Thus, any significant change in the neuronal firing rate recorded after MPD injection can be attributed to the effect of MPD.

### 2.8. Effect of 0.6 mg/kg MPD on VTA, LC, and DR Neurons Recorded from All Group (Table 2 and Figure 3, Upper Panel)

A total of 141 VTA, 134 LC, and 137 DR neurons were recorded from the all animals group receiving 0.6 mg/kg MPD (Table 3). Figure 3 summarizes the total responsiveness for the VTA, LC, and DR neuronal responses to 0.6, 2.5, and 10.0 mg/kg MPD and shows the percentage (%) responsiveness (of how many neurons) following each acute and chronic MPD dose (Figure 3, 0.6 mg/kg MPD). A total of 28.3%, 61.9%, and % and 57.8% of VTA, LC, and DR responded by changing their firing rate to acute 0.6 mg/kg MPD (Table 2, 0.6 mg/kg and Figure 3 (number 1—above)).

**Figure 3 ijms-24-16628-f003:**
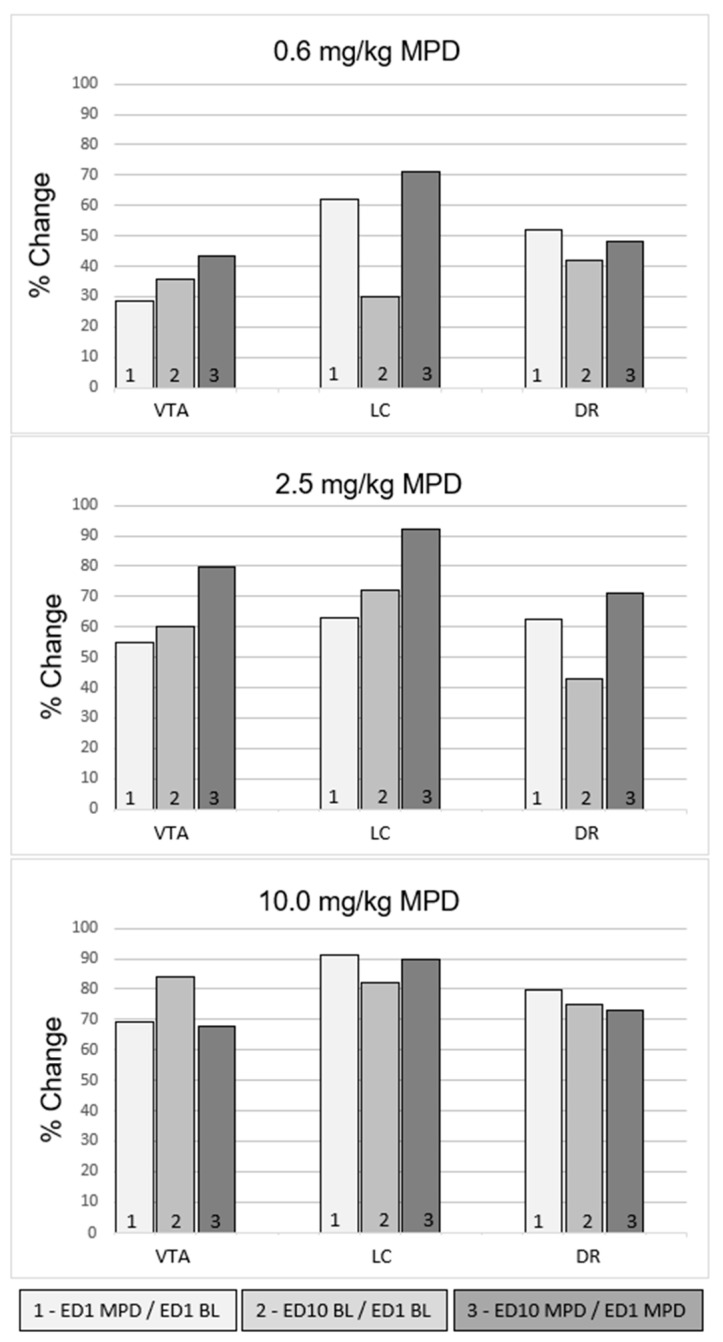
Summary of the statistically (*p* < 0.05) calculated neuronal responses percentages (%) of VTA, LC, and DR neurons of the all animals group. The figure demonstrates (in %) how many VTA, LC, and DR neurons responded to MPD. Each brain structure has a significantly (*p* < 0.05) different response, which was calculated using the Chi-squared test.

When comparing ED10 BL activity to ED1 BL (ED10 BL/ED1 BL) at 0.6 mg/kg MPD treatment (Figure 3 (number 2—above)), 35.5% of VTA neurons, 29.9% of LC neurons, and 41.8% of VTA, LC, and DR neurons displayed a significant (*p* < 0.05) change in neuronal activity, respectively. This change in firing rates after six daily MPD exposures and three washout days expresses neurophysiological withdrawal.

When comparing the chronic effect of MPD (ED10 MPD/ED1 MPD), 43.3% of VTA neurons, 70.9% of LC neurons, and 48.2% of DR neurons displayed a significant (*p* < 0.05) change in neuronal activity.

### 2.9. Effect of 2.5 mg/kg MPD on VTA, LC, and DR Neuronal Units Recorded in All Group (Table 2 and Figure 3, Middle Panels)

A total of 142 VTA, 146 LC, and 142 DR neuronal units were recorded from all animals receiving 2.5 mg/kg MPD (Table 2). A total of 54.9% of VTA neurons, 63% of LC neuronal units, and 51.8% of DR neuronal units demonstrated a significant (*p* < 0.05) change in the neuronal firing rate in response to the acute 2.5 mg/kg MPD effect (ED1 MPD/ED1 BL) (Figure 3, 2.5 mg/kg MPD (number 1—above)).

When comparing ED10 BL neuronal activity after six daily 2.5 mg/kg MPD and three washout days to ED1 BL (ED10 BL/ED1 BL), 59.9% of VTA neuronal units, 71.9% of LC neurons, and 43% of DR neuronal units, respectfully, displayed a significant (*p* < 0.05) change in their neuronal activity. These significant (*p* < 0.05) changes express electrophysiological withdrawal (Figure 3, 2.5 mg/kg MPD (number 2—above)).

When comparing the chronic effect of MPD (ED10 MPD/ED1 MPD), 79.6% of VTA neurons, 52.1% of LC neurons, and 73.8% of DR neurons displayed a significant (*p* < 0.05) change in neuronal activity (Figure 3, 2.5 mg/kg MPD (number 3—above)).

### 2.10. Effect of 10.0 mg/kg MPD on VTA, LC, and DR Neuronal Units Recorded in All Group (Table 2 and Figure 3, Lower Panels)

A total of 152 VTA, 140 LC, and 168 DR neurons were recorded from all animals receiving 10.0 mg/kg MPD (Table 2). A total of 69% of VTA neurons, 94.4% of LC neurons, and 79.8% of DR neurons demonstrated a significant (*p* < 0.05) change in the neuronal firing rate in response to the acute 10.0 mg/kg MPD effect (ED1 MPD/ED1 BL), (Figure 3, 10.0 mg/kg MPD (number 1—above).

When comparing ED10 BL activity to ED1 BL (ED10 BL/ED1 BL), 84.2% of VTA neurons, 92.1% of LC neurons, and 75% of DR neurons displayed a significant (*p* < 0.05) change in activity.

When comparing the chronic effect of MPD to the acute effect of MPD (ED10 MPD/ED1 MPD), 67.8% of VTA neurons, 90% of LC neurons, and 73.8% of DR neurons displayed a significant (*p* < 0.05) change in their neuronal activity.

Statistical comparisons of VTA, LC, and DR neuronal units were recorded from all animals (Figure 3).

After acute and chronic 0.6 mg/kg MPD exposure, the percentage of the neuronal responses recorded from VTA, LC, and DR were significantly different (df: 2; *χ*^2^ = 13.42; *p* < 0.05 df: 2; *χ*^2^ = 14.04; *p* < 0.05) from one another (Figure 3, upper histograms). Using the post hoc comparisons of the neuronal firing rate between the percentage responses to 0.6 mg/kg MPD from the VTA, LC, and DR brain regions revealed that the percentage responses between the three brain areas were significantly (*p* < 0.05) different.

After acute and chronic 2.5 mg/kg MPD exposure, MPD elicited significantly different (df: 2; *χ*^2^ = 10.04; *p* < 0.05 and df: 2; *χ*^2^ = 15.74; *p* < 0.05) neuronal responses between the three brain regions in all animals. Post hoc comparisons of the neuronal firing rate between individual brain regions again revealed that the increase in percentage proportion of the VTA neuron firing rate in response to MPD was significantly (*p* < 0.05) different when compared individually to LC and DR neurons, respectively (Figure 3, middle histograms).

After acute and chronic 10.0 mg/kg MPD exposure, the percentage of the neuronal responses recorded from the VTA, LC, and DR were significantly different (df: 2; *χ*^2^ = 11.33; *p* < 0.05 and df: 2; *χ*^2^ = 13.82; *p* < 0.05) from one another (Figure 3, lower histograms). Post hoc comparisons of the neuronal firing rate between percentage responses from the VTA, LC, and DR brain regions neurons revealed that the percentage responses between the three brain areas were significantly different (*p* < 0.05) (Figure 3, lower histograms).

The ED10 BL/ED1 BL, after six daily MPD exposures of 0.6, 2.5, and 10.0 mg/kg, was significantly different (df: 2; *χ*^2^ = 47.32; *p* < 0.05) between the three brain regions (Figure 3 (number 2—above). Post hoc comparisons of the neuronal firing rate between percentage responses from the VTA, LC, and DR brain regions revealed that the percentage responses between the three brain areas were significantly different (*p* < 0.05) (Figure 3, lower histograms (number 2—above)).

Effect of 0.6 mg/kg MPD on the VTA, LC, and DR neuronal response direction recorded in animals that exhibited behavioral sensitization (Figure 4).

A total of 29% of VTA neurons, 64% of LC neurons, and 68% of DR neurons as well as 71% VTA, 36% LC, and 32% DR neurons were recorded from behaviorally sensitized animals exhibiting a significant (*p* < 0.05) increase or decrease, respectfully, following acute 0.6 mg/kg MPD administration (Figure 4). The neuronal response direction (i.e., increase/decrease) was significantly different (df: 2; *χ*^2^ = 11.7; *p* < 0.05) comparing VTA vs. LC vs. DR, respectively (Figure 4, upper left, 0.6 mg/kg MPD, histograms).

When comparing ED10 BL activity to ED1 BL (ED10 BL/ED1 BL), 71% of VTA neurons, 41% of LC neurons, and 55% of DR neurons displayed a significant (*p* < 0.05) increase in ED10 BL after six daily MPD exposures and three washout days, and 29% VTA neurons, 59% LC neurons, and 45% DR neurons displayed a significant (*p* < 0.05) decrease in their ED10 BL/ED1 BL, respectively. Significant (df: 2; *χ*^2^ = 14.56; *p* < 0.05) differences in how the ED10 BL changed in comparison to VTA neurons vs. LC neurons vs. DR neurons, respectively, were observed (Figure 4, middle histograms, 0.6 mg/kg/MPD).

When comparing the chronic effect of 0.6 mg/kg MPD (ED10 MPD/ED1 MPD), 82% of VTA neurons, 88% of LC neurons, and 66% of DR neurons displayed a significant (*p* < 0.05) increase in neuronal activity, and 18% VTA, 12% LC, and 34% DR neurons displayed a significant (*p* < 0.05) decrease in their neuronal firing rates, respectively. Significant differences (df: 2; *χ*^2^ = 15.42; *p* < 0.05) were observed between VTA and DR, and between LC and DR neuronal response directions (Figure 4, 0.6 mg/kg MPD, histograms).

### 2.11. Effect of 2.5 mg/kg MPD on VTA, LC, and DR Neuronal Response Directions Recorded in Animals That Exhibited Behavioral Sensitization (Figure 4)

A total of 68% VTA, 64% LC, and 89% DR neurons, and 32% VTA, 36% LC, and 16% DR neurons were recorded from behaviorally sensitized animals receiving acute 2.5 mg/kg MPD that exhibited a significant (*p* < 0.05) increase or decrease in their firing rates, respectively, in response to acute 2.5 mg/kg MPD. The neuronal response direction was significantly different (df: 2; *χ*^2^ = 14.23; *p* < 0.05) comparing VTA to LC and DR (Figure 4, 2.5 mg/kg MPD, histograms).

When comparing ED10 BL activity to ED1 BL (ED10 BL/ED1 BL), 72% of VTA neurons, 75% of LC neurons, and 80% of DR neurons displayed a significant (*p* < 0.05) increase in neuronal activity, and 28% of VTA, 25% of LC, and 20% of DR neurons displayed a significant (*p* < 0.05) decrease in their neuronal activities after six daily 2.5 mg/kg MPD treatments and three washout days, respectively. No significant differences were observed in ED10 BL/ED1 BL between the VTA, LC, and DR neuronal activities (Figure 4, 2.5 mg/kg MPD, histograms).

When comparing the chronic effect of chronic 2.5 mg/kg MPD (ED10 MPD/ED1 MPD), 76% of VTA neurons, 70% of LC neurons, and 81% of DR neurons displayed a significant (*p* < 0.05) increase in neuronal activity, and 24% of VTA, 30% of LC, and 19% of DR neurons displayed a significant (*p* < 0.05) decrease in firing rates. Significant differences (df: 2; *χ*^2^ = 10.30; *p* < 0.05) were observed when comparing the response directions between VTA vs. DR and LC vs. DR neuronal responses to chronic 2.5 mg/kg MPD (Figure 4, lower, 2.5 mg/kg MPD).

### 2.12. Effect of 10.0 mg/kg MPD on VTA, LC, and DR Neuronal Response Directions Recorded in Animals That Exhibited Behavioral Sensitization (Figure 4)

Ninety-five percent of VTA, 85% of LC, and 71% of DR neurons; and 5% of VTA, 15% of LC and 29% of DR neurons were recorded from behaviorally sensitized animals receiving 10.0 mg/kg MPD exhibiting significant (*p* < 0.05) increase or decrease in their firing rates, respectively following acute 10 mg/kg MPD. The neuronal response direction was significantly different (df: 2; *χ*^2^ = 15.30; *p* < 0.05) comparing the VTA, LC and DR neuronal response directions (Figure 4, 10.0 mg/kg MPD, histograms).

When comparing ED10 BL activity to ED1 BL (ED10 BL/ED1 BL) in sensitized animals, 72% of VTA neurons, 88% of LC neurons, and 59% of DR neurons displayed a significant (*p* < 0.05) increase in their neuronal firing rate, and 28% of VTA, 12% of LC and 41% of DR neuronal activity displayed a significant (*p* < 0.05) increase or decrease, respectively. The neuronal response direction of ED10 BL/ED1 BL was significantly different (df: 2; *χ*^2^ = 10.92; *p* < 0.05) comparing the VTA, LC and DR neuronal firing rates (Figure 4, 10.0 mg/kg MPD, histograms).

When comparing the chronic effect of 10.0 mg/kg MPD (ED10 MPD/ED1 MPD) in sensitized animals, 62% of VTA neurons 84% of LC neurons, and 66% of DR neurons displayed a significant (*p* < 0.05) increase in neuronal activity, and 38% of VTA, 16% of LC and 34% of DR neuronal responses displayed significant (*p* < 0.05) decreases, respectively. Significant differences (df: 2; *χ*^2^ = 14.77; *p* < 0.05) were observed between the above three brain area response directions to ED10 MPD/ED1 MPD, respectively (Figure 4, 10.0 mg/kg MPD, histograms).

### 2.13. Effect of 0.6 mg/kg MPD on VTA, LC, and DR Neuronal Response Directions Recorded in Animals That Exhibited Behavioral Tolerance (Figure 5)

A total of 47% of VTA neurons, 27% of LC neurons, and 45% of DR neurons as well as 53% VTA, 73% LC, and 55% DR neurons were recorded from behaviorally tolerant animals exhibiting a significant (*p* < 0.05) increase or decrease, respectfully, following acute 0.6 mg/kg MPD (Figure 5). The neuronal response direction (i.e., increase/decrease) was significantly different (df: 2; *χ*^2^ = 11.7; *p* < 0.05) when comparing VTA, vs. LC vs. DR, respectively (Figure 5, 0.6 mg/kg, histograms).

**Figure 5 ijms-24-16628-f005:**
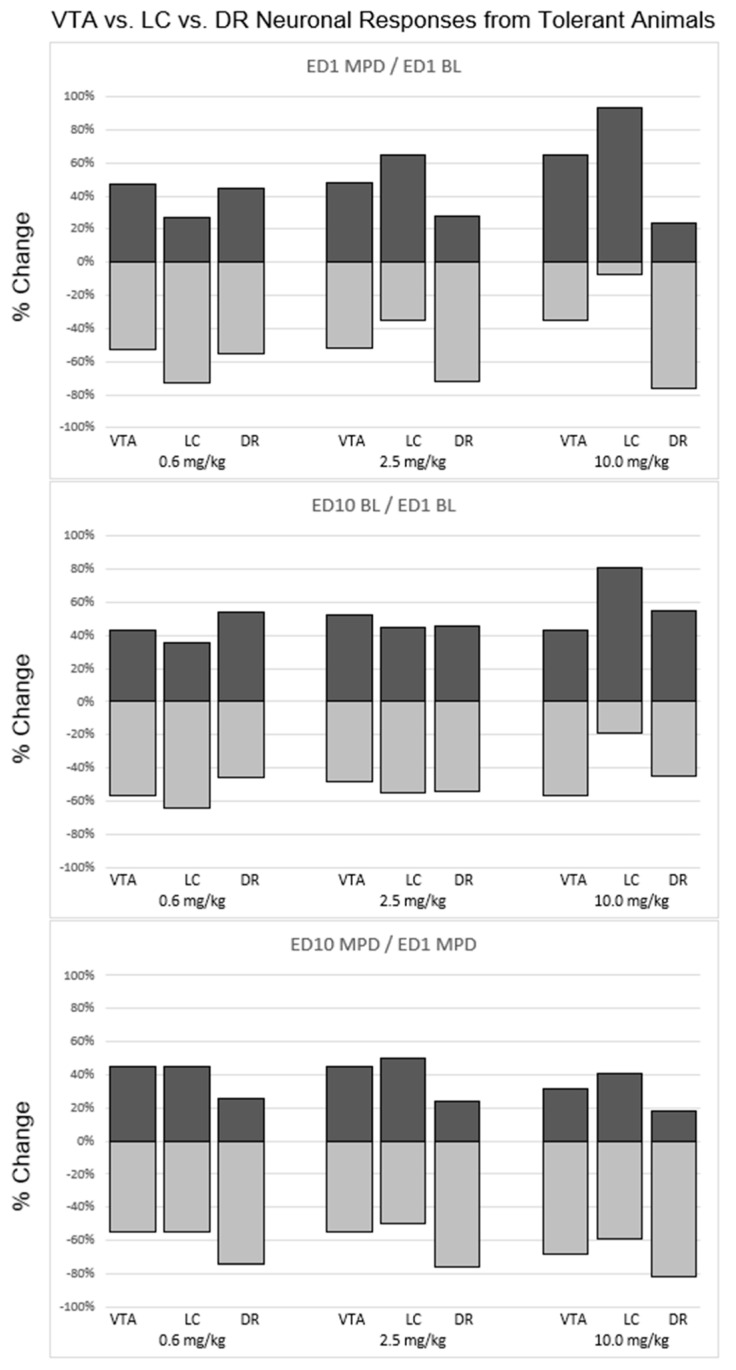
Summary of the statistically calculated direction (excitation/attenuation) neuronal responses of VTA, LC, and DR neurons recorded from animals that exhibited behavioral tolerance after chronic MPD exposure. Each bar represents a percentage of the neuronal responses within each brain region that responded to each respective MPD dose. A positive percentage represents the proportion of neurons within a brain region that exhibited a significant (*p* < 0.05) increase in neuronal activity. A negative percentage represents the proportion of neurons within a brain region that exhibited a significant (*p* < 0.05) decrease in neuronal activity. Top panel: comparison of VTA, LC, and DR responses after acute MPD exposure (ED1 MPD/ED1 BL). Middle panel: comparison of VTA, LC, and DR responses at baseline (ED10 BL/ED1 BL). Bottom panel: comparison of VTA, LC, and DR responses after chronic MPD exposure (ED10 MPD/ED1 MPD).

When comparing ED10 BL activity to ED1 BL (ED10 BL/ED1 BL), 43% of VTA neurons, 36% of LC neurons, and 54% of DR neurons displayed a significant (*p* < 0.05) increase in ED10 BL after six daily MPD exposures and three washout days, and 57% VTA neurons, 64% LC neurons, and 46% DR neurons displayed a significant (*p* < 0.05) decrease in their ED10 BL/ED1 BL, respectively. There were also significant differences (df: 2; *χ*^2^ = 11.56; *p* < 0.05) in the ED10 BL/ED1 BL when comparing VTA neurons vs. LC neurons vs. DR neural activities, respectively (Figure 5, 0.6 mg/kg/MPD).

When comparing the chronic effect of 0.6 mg/kg MPD (ED10 MPD/ED1 MPD), 45% of VTA neurons, 45% of LC neurons, and 26% of DR neurons displayed a significant (*p* < 0.05) increase in neuronal activity, and 55% VTA, 55% LC, and 74% DR neurons displayed a significant (*p* < 0.05) decrease in their neuronal firing rates, respectively. Significant differences (df: 2; *χ*^2^ = 10.42; *p* < 0.05) were observed between VTA and DR, and between LC and DR neuronal response directions (Figure 5, 0.6 mg/kg MPD, histograms).

### 2.14. Effect of 2.5 mg/kg MPD on VTA, LC, and DR Neuronal Response Directions Recorded in Animals That Exhibited Behavioral Tolerance (Figure 5)

A total of 48% VTA, 65% LC, and 28% DR neurons, and 52% VTA, 35% LC, and 12% DR neurons were recorded from behaviorally tolerant animals receiving acute 2.5 mg/kg MPD that exhibited a significant (*p* < 0.05) increase or decrease in their firing rates, respectively, to acute 2.5 mg/kg MPD. The neuronal response directions were significantly different (df: 2; *χ*^2^ = 14.23; *p* < 0.05) when comparing VTA to LC and DR (Figure 5, 2.5 mg/kg MPD, histograms).

When comparing ED10 BL activity to ED1 BL (ED10 BL/ED1 BL), 52% of VTA neurons, 45% of LC neurons, and 46% of DR neurons displayed a significant (*p* < 0.05) increase in neuronal activity, and 48% of VTA, 55% of LC, and 54% of DR neurons displayed a significant (*p* < 0.05) decrease in neuronal activity after six daily 2.5 mg/kg MPD treatments and three washout days, respectively. No significant differences were observed in ED10 BL/ED1 BL between the VTA, LC, and DR neuronal activities (Figure 5, 2.5 mg/kg MPD, histograms).

When comparing the chronic effect of 2.5 mg/kg MPD (ED10 MPD/ED1 MPD), 45% of VTA neurons, 50% of LC neurons, and 24% of DR neurons displayed a significant (*p* < 0.05) increase in neuronal activity, and 55% of VTA, 50% of LC, and 76% of DR neurons displayed a significant (*p* < 0.05) decrease in firing rates. Significant differences (df: 2; *χ*^2^ = 10.30; *p* < 0.05) were observed when comparing the response directions between VTA vs. DR and LC vs. DR neuronal responses to chronic 2.5 mg/kg MPD (Figure 5, histograms, 2.5 mg/kg MPD).

### 2.15. Effect of 10.0 mg/kg MPD on VTA, LC, and DR Neuronal Response Directions Recorded in Animals That Exhibited Behavioral Tolerance (Figure 5)

A total of 65% of VTA, 93% of LC, and 24% of DR neurons, and 35% of VTA, 7% of LC, and 76% of DR neurons were recorded from behaviorally tolerance animals receiving 10.0 mg/kg MPD that exhibited a significant (*p* < 0.05) increase or decrease in their firing rates, respectively, following acute 10.0 mg/MPD. The neuronal response direction was significantly different (df: 2; *χ*^2^ = 15.30; *p* < 0.05) when comparing the VTA, LC, and DR neuronal response directions (Figure 5, 10.0 mg/kg MPD, upper histograms).

When comparing the ED10 BL activity to ED1 BL (ED10 BL/ED1 BL) in behaviorally tolerant animals, 43% of VTA neurons, 81% of LC neurons, and 55% of DR neurons displayed a significant (*p* < 0.05) increase in their neuronal activity, and 57% of VTA, 19% of LC, and 45% of DR neuronal activities displayed significant (*p* < 0.05) decreases, respectively. The neuronal response directions of ED10 BL/ED1 BL were significantly different (df: 2; *χ*^2^ = 10.95; *p* < 0.05) when comparing the VTA, LC, and DR neuronal firing rates (Figure 5, 10.0 mg/kg MPD, histograms).

When comparing the chronic effect of 10.0 mg/kg MPD (ED10 MPD/ED1 MPD) in behaviorally tolerant animals, 32% of VTA neurons, 41% of LC neurons, and 18% of DR neurons displayed a significant (*p* < 0.05) increase in neuronal activity, and 68% of VTA, 59% of LC, and 82% of DR neuronal responses displayed significant (*p* < 0.05) decreases, respectively. Significant differences (df: 2; *χ*^2^ = 14.77; *p* < 0.05) were observed between the above three brain area response directions to ED10 MPD/ED1 MPD, respectively (Figure 5, 10.0 mg/kg MPD, histograms).

## 3. Discussion

Methylphenidate (MPD) exerts its effects mainly by binding to the dopamine (DA), norepinephrine (NE), and serotonin (5HT) transporters in the synaptic cleft, preventing their reuptake back into the presynaptic terminal, resulting in increased concentrations of DA, NE, and 5HT within the synaptic cleft [24,25,27]. The ventral tegmental area (VTA), a group of neurons located on the floor of the midbrain, is the primary source of dopamine synthesis that projects into the mesocorticolimbic pathway and is responsible for its many functions. The VTA is composed of dopaminergic, glutamatergic, as well as GABAergic neurons, making it a heterogenous group of neurons. The medial VTA participates in an inhibitory pathway, while the lateral VTA participates predominantly in the dopaminergic reward pathway [32,33,34]. DA signaling participates in the regulation of attention, cognition, and motor activity [31,35]. The locus coeruleus (LC) are a group of neurons located deep within the brainstem and are the origin of synthesis of NE. This region contains many projections that provide far-reaching noradrenergic transmissions throughout the brain [35,36,37]. Noradrenergic signaling within the cortical and subcortical structures regulate attention, memory, and sensation, and the LC is directly involved in the alteration of gene expression that determines executive function, behavior, and arousal [38,39,40,41,42]. The dorsal raphe (DR) nucleus, located in the brainstem, contains serotonergic (5-HT) neurons [27,43,44]. 5-HT produced in the DR has been shown to exhibit complex interactions with other monoamines, such as DA and NE, and has been shown to result in increased action impulsivity and failure to wait for delayed rewards (Miyazaki et al., 2012 [43]).

The objective of this study was to investigate the role of MPD on the VTA, LC, and DR neurons of young animals whose brains are still going through neuronal development [13]. The main findings of the behavioral observation are that each MPD dose (0.6, 2.5, 10.0 mg/kg) elicits behavioral sensitization in some animals, and displays behavioral tolerance in others, as compared to the initial effect of the drug (Figure 2). Due to the observation that the same dose of MPD elicits behavioral sensitization in some animals and behavioral tolerance in others, we also evaluated the neuronal activity based on the behavioral response to chronic MPD exposure from each of the three brain structures, VTA, LC, and DR. Neuronal recordings obtained from animals expressing behavioral sensitization to chronic MPD mainly exhibit excitation, when compared to the initial effects of MPD. Neuronal recordings obtained from animals expressing behavioral tolerance to chronic MPD exposure mainly exhibited attenuations in relation to the initial effect of MPD. In general, the neuronal recordings’ percentage responsiveness and the response directions of the responses from the VTA, LC, and DR from the animals expressing behavioral sensitization were significantly different from the neuronal recordings’ percentage responsiveness obtained from the animals expressing behavioral tolerance to chronic MPD exposure, respectively.

### 3.1. Acute MPD (ED1 MPD/ED1 BL) on Total Responsiveness

When increasing the MPD dose, more neurons responded to the drug. Significant differences in the percentages of how many neurons respond were observed (Figure 3) following acute 0.6 mg/kg MPD administration. The LC neurons’ percentage responses to MPD were the highest, followed by the DR neurons and then the VTA neurons (Figure 3 (number 1—above). Following the administration of 2.5 and 10.0 mg/kg MPD, different total responsiveness was obtained. That is, each of the VTA, LC, or DR neurons respond differently to the acute MPD.

### 3.2. Baseline Activity (ED10 BL/ED1 BL) as a Result of Chronically (Repetitively) Increasing the MPD Dose

After six daily injections of 0.6, 2.5, and 10.0 mg/kg MPD and three washout days, the ED10 BL/ED1 BL neuronal activity exhibited a different dose response effect in the neuronal recordings from the three brain regions, suggesting that the three brain areas participated differently in the regulation of MPD withdrawal behavior (Figure 2 (number 2—above)).

### 3.3. Chronic MPD (ED10 MPD/ED1 MPD) Effect on Total Responsiveness

Each of the VTA, LC, and DR neurons’ total percentage responsiveness showed different percentages for their neuronal responses to each of the MPD doses tested (Figure 3 above number 3), suggesting that each brain area exhibits different roles in response to MPD and responds differently to chronic MPD exposure.

The observation that each brain area responds differently to MPD agrees with the observations of the study in humans which used positron emission tomography (PET). Studies in the literature report differences in the brain concentrations of DA, NE, and 5HT transporters [45,46]. Following MPD exposure, a 2200-times greater affinity to DAT and 1300-times greater affinity to NAT was observed in comparison to SERT [47], while the effect of MPD on the neuronal activity firing rate in the VTA, LC, and DR did not follow the same levels of the above transporter concentrations in these areas, respectively.

### 3.4. Acute and Chronic MPD on Neuronal Response Directions (Excitation/Attenuation)

The observations following acute and chronic 0.6, 2.5, and 10.0 mg/kg MPD within and between each of the three brain areas exhibited different ratios of the number of neurons in the VTA, LC, or DR that responded via excitation to attenuation (Figure 4 and Figure 5). Significant differences in the response directions between the recordings obtained from behaviorally sensitized to behaviorally tolerant neurons were observed (Figure 4 vs. Figure 5). These different response directions suggest that each of the VTA, LC, and DR areas has a different role to each of the acute or chronic 0.6, 2.5, and 10.0 mg/kg MPD and that DA, NE, and 5HT participate differently in relation to the effect of MPD.

MPD acts primarily as the DA, NE, and 5HT reuptake inhibitor; at the same time, it secures the balance between the availability of intra-synaptic DA, NE, and 5HT, and the intracellular pool of neurotransmitters through the interaction with vesicular monoamine transporter 2 (VMAT2) [48]. In monoaminergic neurons, it was observed that MPD affects the redistribution of VMAT2, which is involved in the sequestration of cytoplasmic DA, NE, and 5HT in each brain area differently, and is an important regulator of neurotransmission [20].

How should this observation be interpreted? We interpreted that there are two dichotomous mechanisms in each brain site, excitation and inhibition, in response to any stimulus, which is MPD in this case. The ratio of this excitation–inhibition mechanism is essential to how a particular brain area will respond to a certain stimulus, and the ratio between these two dichotomous responses in each brain structure determines the role of each brain area in response to a given stimulus. Thus, the different roles of each brain area in response to a drug are determined by the ratio how many neurons respond via excitation to attenuation in that specific region. As different observations were made in each of the three brain areas after the administration of the 0.6 and 2.5 mg/kg MPD doses, this suggests that each structure responds in its own unique way to varying MPD doses, as opposed to acting as one large unit with the same responses throughout.

In conclusion, this study strongly proposes the importance of concomitantly studying the behavioral and neuronal responses from several brain areas simultaneously before and following the dose responses of acute and chronic psychostimulant exposure to obtain an accurate role of the drug which is under investigation.

## 4. Materials and Methods

### 4.1. Animals

Young male Sprague Dawley rats at post-natal day 30 were purchased (Harlan, Indianapolis, IN, USA) and allowed 3–5 days of acclimation prior to bilateral VTA, LC, and DR recording electrode implantation. Food and water were provided ad libitum. Room temperature was maintained at 21 ± 2 °C with a relative humidity of 37–45% under 12 h:12 h alternating light–dark cycle with lights on at 6:00. After electrode implantation, animals were returned to their home cages, which were also used as the test cages for the duration of the electrophysiological and behavioral recordings for an additional 5–7 days prior to the neuronal and behavioral recording session days. All experimental procedures were approved by the University of Texas Health Science Center Animal Welfare Committee and were performed in accordance to the National Institute of Health Guide for Care and Use of Laboratory Animals.

### 4.2. Surgery

Young animals were anesthetized with a 30 mg/kg pentobarbital intraperitoneal (i.p) injection. Each rat’s head was shaved, and lidocaine hydrochloride topical gel was applied to the shaved area for local anesthetic. The animal was then placed in a stereotaxic head holder instrument, and an incision was made to expose the skull. Bilateral holes were drilled on the skull above the VTA posterior (P) from Bregma 4.7 mm; lateral (L) from midline 0.6 mm, LC P 8.0 mm; L 1.2 mm and DR at P 7.0 mm; L 0.2 mm using the Sherwood and Timiras Young rat brain atlas [49], with an additional hole in front of the frontal sinus for the reference electrode. Six anchor screws were inserted in the vacant spots of the skull to secure the skull cap with dental acrylic cement. Two nickel–chromium wires (insulated, except at tips) 60 µm in diameter were twisted, and each wire was secured to 1 cm copper connector pins and individually inserted bilaterally into the VTA, LC, and DR, such that each animal had 12 total recording electrodes and a ground electrode. During placement of the electrodes, the neuronal unit activity was monitored. The electrodes were inserted at a depth of 7.0 mm, 7.0 mm, and 6.2 mm in the VTA, LC, and DR, respectively. If sufficient neuronal activity was observed, the wires were secured to the anchor screws and to the skull with dental acrylic cement. Electrodes failing to detect satisfactory activity were lowered in increments of 5 to 10 µm until satisfactory spike activity demonstrated a 3:1 signal to noise ratio [13,18,19]. Animals were allowed to recover from the surgical procedure for 5 to 7 days. During recovery, animals were placed within their home cage in the experimental room for two hours each day and connected to the wireless head stage (Triangle Biosystems Inc, TBSI, Durham, NC, USA) to adapt and acclimate to the neuronal and behavioral recording systems.

### 4.3. Experimental Protocol and Data Acquisition

The neuronal and locomotor behavioral activity of the animals started jointly at age P-40 (Table 1) using a wireless neuronal recording system (TBSI, Durham, NC, USA) and an open field computerized animal activity system (Accuscan, Columbus, OH, USA). The TBSI head stage was connected to the rat head cap containing the electrode pins and sent electrical signals through a transmitter to a remote receiver connected to an analog-to-digital converter (Micro 1401-3; Cambridge Electronic Design (CED)). The neuronal activity from each electrode was collected and stored on a PC using CED Spike 2.7 software.

The open field system consisted of a 40 cm × 40 cm × 32 cm cage with 16 × 16 infrared beams, and their sensors were placed 5, 8, and 12 cm above the floor of the cage in the opposite side (Figure 6). This setup has been previously described in detail ([18,19]). Interruption of the infrared beams by animal movement was detected by the monitoring system at 100 Hz frequency and the interruptions were compiled with Oasis software (Accuscan, Columbus, OH, USA) and downloaded to a PC every 10 min. The software categorized beam interruptions into three different locomotive behaviors: number of movements (NM), total distance (TD) traveled in cm, and number of stereotypic movements (NOS). This is a count of repetitive movements with at least one-second intervals between movements activating the same sensor (Figure 6). Data were recorded for 60 min post-injection of either saline or MPD on experimental day (ED) 1 and ED10. The behavioral recording was performed to provide a way to distinguish between animals that exhibited behavioral sensitization following repetitive MPD exposure from animals that exhibited behavioral tolerance as compared to the initial MPD effect, respectively [13]. The behavioral locomotive data were used as the basis for analysis of neuronal recordings, since some animals exhibited behavioral sensitization and others behavioral tolerance to each of the 0.6, 2.5, and 10.0 mg/kg doses of MPD groups. The data evaluations for both the behavioral and the neuronal recording were divided into three groups: (1) data obtained from all animals—all group; (2) data obtained only from animals expressing behavioral sensitization—sensitized group; and (3) data obtained only from animals expressing behavioral tolerance—tolerant group. Recording started at post-natal age of 40 days (P-40) and lasted for 10 consecutive days (Table 1). The rats were randomly subdivided into four groups: saline 0.8 cc (control), and 0.6, 2.5, and 10.0 mg/kg MPD treatment groups. On experimental day 1 (ED1), rats were placed within their home cage in a Faraday testing cage to reduce background noise. The wireless head stage was connected to the electrode pins of the skull cap, and animals were allowed to acclimate for an additional 20 to 30 min prior to the recording session. On ED1, neuronal and behavioral activity was recorded concomitantly for one hour following an initial injection of 0.8 mL saline; this information serves as the baseline activity (ED1 BL). The animals then received a second injection of either saline (saline/saline group) or 0.6, 2.5, or 10.0 mg/kg MPD, and neuronal and locomotive behavior was recorded for one additional hour (ED1 BL or ED1 MPD, respectively). From ED2 through ED6, animals received either saline or MPD injections in their home cages without behavioral or neuronal recordings in order to induce chronic MPD effects. From ED7 through ED9, the animals underwent a three-day washout period where no injections were administered. On ED10, animals were administered a 0.8 mL saline injection, and baseline neuronal and behavioral activity was recorded for one hour (ED10 BL). The animals were then rechallenged with either saline or 0.6, 2.5, or 10.0 mg/kg MPD (ED10 BL or ED10 MPD, respectively), and an additional hour of recording was performed, as on ED1 (Table 1) [13,18,19].

### 4.4. Drug

Methylphenidate hydrochloride (MPD) was donated by Mallinckrodt (St. Louis, MO, USA). Previous experiments using dose response MPD protocols from 0.1 to 40.0 mg/kg i.p have found that behavioral effects of MPD were observed from 0.6 mg/kg doses MPD and higher [50]. As such, MPD was administered at 0.6, 2.5, and 10.0 mg/kg corresponding to low, moderate, and high experimental dosages, respectively. MPD was dissolved in a 0.9% isotonic saline solution for i.p injection. Control subjects received injections of 0.8 mL isotonic saline solution (0.9% NaCl). All MPD injections were titrated to a volume of 0.8 mL with 0.9% saline to equalize MPD injection volumes for all animals.

### 4.5. Histological Verification of Electrode Placement

An overdose of sodium phenobarbital was administered upon completion of recording on ED11. Animals were perfused intracardially with 10% formaldehyde solution containing 3% potassium ferrocyanide and a 20 µA current was passed through each electrode pin for 20 s to produce a small lesion at the recording sites. The brain was removed and preserved in 10% formalin for histological processing. The positions of the electrodes in the VTA, LC, and DR were confirmed by the location of the lesion and Prussian blue spot using the Adolescent Rat Brain Atlas [49].

### 4.6. Behavioral Data Analysis

The data acquired were used to make three comparisons: (1) locomotor and neuronal activity after MPD administration on ED1 was compared to the baseline (BL) recording post-saline administration on ED1 (ED1 MPD/ED1 BL) in order to determine the acute effects of MPD; (2) activity (behavioral and neuronal) post-saline injection on ED10 was compared to post-saline injection activity on ED1 (ED10 BL/ED1 BL) in order to determine any significant changes in BL activity after six daily repetitive (chronic) MPD exposures and three washout days; and (3) activity after MPD administration on ED10 was compared to activity after MPD administration on ED1 (ED10 MPD/ED1 MPD) in order to determine the chronic effect of MPD and to find out whether the animal displayed behavioral sensitization or tolerance. Animals displaying significantly increased locomotor activity after MPD rechallenge on ED10 as compared to MPD administration on ED1 (ED10 MPD/ED1 MPD) were considered to display behavioral sensitization, while those exhibiting a significant decrease in locomotor activity following MPD rechallenge on ED10 as compared to MPD administration on ED1 were considered to display behavioral tolerance. Student’s *t*-test and the critical ratio (*C.R.*) test were used to determine the effect of MPD on individual animals [13,27].
$$C.R.=\frac{E-C}{\sqrt{E+C}}=\pm 1.96\;=\;p\;<\;0.05$$

For acute effect of MPD, *E* represents activity count after MPD injection on ED1 and *C* represents the ED1 activity after saline (control) injection (ED1 MPD/ED1 BL). For multiple injections (chronic) effect, *E* represents activity following Sal or MPD injection on ED10 and *C* represents activity after Sal or MPD dose on ED1 (ED10 BL/ED1 BL and ED10 MPD/ED1 MPD). Individual rats were categorized as exhibiting either behavioral sensitization or tolerance and sorted based on their classification. A significant difference among the groups was determined using a two-way ANOVA, where *p* < 0.05 was accepted as the minimal level of significance.

### 4.7. Neuronal Data Analysis

CED spike 2.7 software was used for spike sorting and statistical analysis of the sorted neuronal data. The data were captured with the program and processed using low- and high-pass filters (0.3–3.3 kHz). There were two window discriminator levels, one for positive-going spikes and one for negative-going spikes (Figure 7). The spikes were extracted when the input signal entered the previously determined amplitude window. Selected spikes with peak amplitudes, which were triggered by the window and exhibited durations of 0.8–1.2 ms, were used to create a template. One thousand waveform data points were used to define the selected spike. The algorithm used to capture a spike allows the extraction of templates that provide high-dimensional reference points that can be used to discriminate consistent and accurate spike sorting despite the influence of some noise, false threshold crossing, and waveform overlap. Incoming spikes were compared with all temporal templates to find the best fitting template that yielded the minimum residue variance. When the distance between the spike waveform and the template exceeded threshold (80%), the waveforms were rejected. This means that the spike sorting accuracy in the reconstructed data was approximately 95%. The same parameters and templates used to sort and count ED1 neuronal activity were loaded onto the ED10 file of the same electrode to sort and count the ED10 activity, ensuring that the waveform sorting parameters used on ED1 and ED10 were identical.

Once spike sorting was completed, the data were exported into a spreadsheet which calculated the average neuronal firing rates for each treatment and produced a sequential firing rates graph (Figure 7). Three statistical comparisons were performed for each neuronal unit as follows: (1) Neuronal unit activity after the initial MPD exposure was compared to neuronal unit activity following saline administration (baseline activity) on ED1 (ED1 MPD/ED1 BL) to obtain the MPD acute effect. (2) Neuronal unit baseline activity on ED10 was compared with the neuronal unit baseline activity on ED1 (ED10 BL/ED1 BL) to obtain whether withdrawal after six daily MPD exposure was expressed. (3) Neuronal unit activity after MPD administration on ED10 was compared to the activity following initial MPD on ED1 (ED10 MPD/ED1 MPD) to obtain the chronic effect of the drug. Significant changes in the neuronal firing rate and direction of change (increase or decrease) for each neuronal unit were determined with Student’s *t*-test and the critical ratio (CR) test. A CR test value above +1.96 indicated that the neuronal unit showed a significant increase in its activity, while a CR value below −1.96 indicated that the unit showed a significant decrease in its activity after MPD administration. In addition, the above neurophysiological data analysis was summarized into three subgroups matched on the basis of an animal’s behavioral response to each chronic dose of MPD treatment as follows: (1) electrophysiological data recorded from all the animals, i.e., the all group; (2) electrophysiological data recorded from animals exhibiting behavioral sensitization, i.e., the sensitized group; and (3) electrophysiological data recorded from animals exhibiting behavioral tolerance, i.e., the tolerance group (Table 2 and Table 3). The significant differences in firing rates of the neuronal units between these three subgroups (all, sensitized, tolerance) was analyzed using the Chi-squared test, where *p* < 0.05 was considered significant. In addition, two statistical methods were used to determine whether MPD exposure caused a significant effect in the neuronal activity of the three different brain regions. To decide whether to use parametric or non-parametric methods, the firing rates were evaluated for normality assumptions. Since the firing rates were determined to not hold normality assumptions, the differences in firing rates were assessed using the statistical critical test, as mentioned above. In this way, the critical ratio test was used to determine if a neuronal unit responded with a significant (*p* < 0.05) increase or decrease in the neuronal firing rate, or no significant change in the neuronal firing rate (Table 2 and Table 3). In addition, in order to evaluate significant differences in neuronal activity within a given brain area or between brain areas (Figure 3, Figure 4 and Figure 5), response ratios were compared. The response ratio refers to the ratio between the number of neuronal units that responded significantly to MPD with an increase in the neuronal firing rate to the number of neuronal units that responded to MPD with a decrease in firing rate for a given subgroup (all animals, sensitized animals or tolerant animals) in relation to a given MPD dose (0.6, 2.5, or 10.0 mg/kg) and a given MPD exposure (acute, baseline, or chronic). The Pearson’s *χ*^2^ test (significance level, *p* < 0.05) was used to evaluate significant differences between response ratios amongst the three brain regions. Post hoc Bonferroni adjustments were performed on groups that showed significant (*p* < 0.05) differences in the *χ*^2^ test to see which of the three brain regions was significantly (Bonferroni adjusted *p* < 0.017) different from the other two.

## Figures and Tables

**Figure 1 ijms-24-16628-f001:**
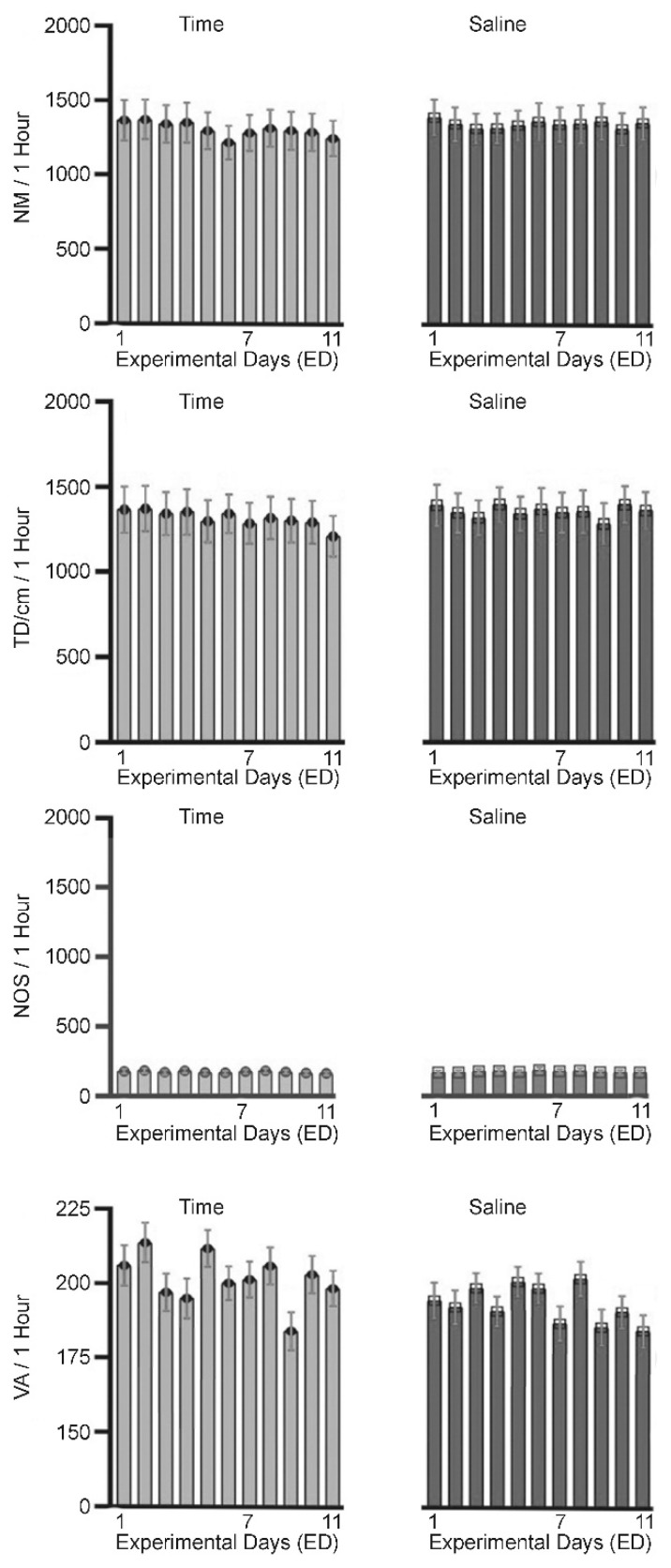
Summary of behavioral activity with no treatment (time (days) control—left column), and the effect of saline injection (see Table 1) on the number of movements (MN), total distance (DT) travelled in cm interrupting the lower panel, number of stereotypic (NOS) movements (interrupting the middle panel), and vertical activity (VA) as result of interrupting the red beam in the third panel.

**Figure 4 ijms-24-16628-f004:**
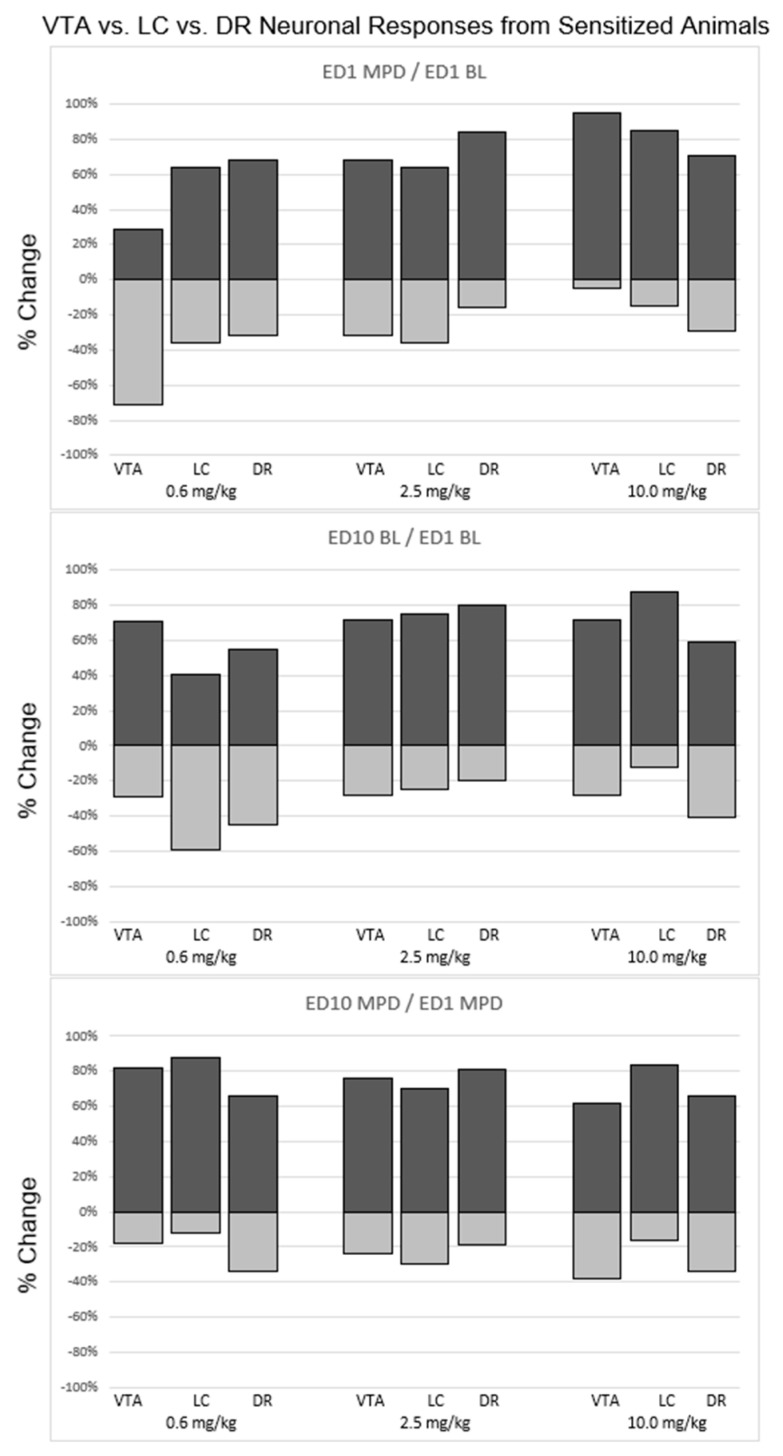
Summary of the statistically calculated direction (excitation/attenuation) of neuronal responses of VTA, LC, and DR neurons recorded from animals that exhibited behavioral sensitization after chronic MPD exposure. Each bar represents a percentage of the neuronal responses within each brain region that responded to each respective MPD dose. A positive percentage represents the proportion of neurons within a brain region that exhibited a significant (*p* < 0.05) increase in neuronal activity. A negative percentage represents the proportion of neurons within a brain region that exhibited a significant (*p* < 0.05) decrease in neuronal activity. (**Top panel**): comparison of VTA, LC, and DR responses after acute MPD exposure (ED1 MPD/ED1 BL). (**Middle panel**): comparison of VTA, LC, and DR responses at baseline (ED10 BL/ED1 BL). (**Bottom panel**): comparison of VTA, LC, and DR responses after chronic MPD exposure (ED10 MPD/ED1 MPD).

**Figure 6 ijms-24-16628-f006:**
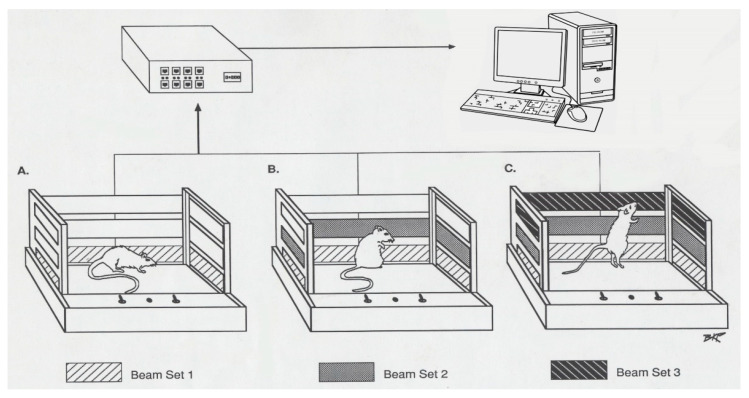
The figure shows three open field cages (A, B, & C) with three different levels of panels (marked in black). Each cage is 40 cm in length, 40 cm in width, and 30 cm in height. Each panel has 16 infra-red beams; on the opposite side, there are their 16 sensors that count the interrupted beams caused by animal movements. The panels are located 4, 8, and 12 cm above the floor of the cage. The lower panels record the number of movements (NM) and the total distance (TD) in centimeters. The second level of panels with sensors records the number of stereotypic movement (NOS), and the third level records the vertical activity (VA) before and after drug exposure.

**Figure 7 ijms-24-16628-f007:**
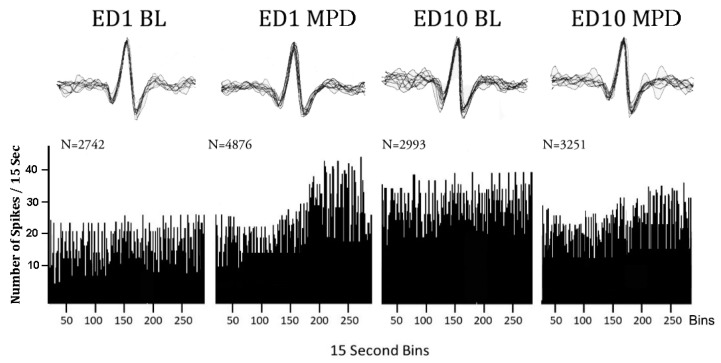
Representative sequential firing rate histograms of VTA neuron before and after 2.5 mg/kg MPD exposure. Left histogram of VTA neuronal activity at baseline on experimental day 1 (ED1 BL). Next histogram of VTA neuronal activity following acute MPD exposure (ED1 MPD). Next histogram of VTA neuronal activity at ED10 BL. Next histogram of VTA neuronal activity following chronic MPD exposure (ED10 MPD). The spike insets above each histogram represent 20 superimposed spikes obtained randomly during the 60 min recordings and show that the same spike patterns and amplitudes were counted during recording sessions. N = total number of VTA neuronal firing rate count/60 min.

**Table 1 ijms-24-16628-t001:** Summary of the experimental protocol for the simultaneous behavioral and neuronal recordings from the respective brain regions of all four treatment groups. On experimental day 1 (ED1), saline was injected and followed by a 60 min behavioral and electrophysiological recording period which served as the baseline (BL). Then, the respective treatments (saline and 0.6, 2.5, or 10.0 mg/kg MPD) were administered and followed by another 60 min recording to obtain the acute effect of the drug. ED2-ED6 consisted of five daily injections of saline or MPD, with a total of six consecutive daily injections including ED1. These daily injections initiated the chronic effect of the drug. ED7-ED9 entails the washout period, during which no injections of any drug were administered. Finally, on ED10, saline was once again administered to all animal groups and followed by a 60 min recording period to establish the ED10 BL after six daily injections and three washout days. This was followed by one last injection of saline or respective MPD doses to assess the chronic effect of MPD, which was subsequently compared to the acute MPD effect obtained after ED1. * Indicates the behavioral and neuronal activity recording days.

		Experimental Days (ED)			
Treatment Groups		ED 1 *	ED 2–6	ED 7–9	ED 10 *
1	Saline	Saline/Saline	Saline	Washout	Saline/Saline
2	0.6 mg/kg MPD	Saline/0.6 mg/kg MPD	0.6 mg/kg MPD	Washout	Saline/0.6 mg/kg MPD
3	2.5 mg/kg MPD	Saline/2.5 mg/kg MPD	2.5 mg/kg MPD	Washout	Saline/2.5 mg/kg MPD
4	10.0 mg/kg MPD	Saline/10.0 mg/kg MPD	10.0 mg/kg MPD	Washout	Saline/10.0 mg/kg MPD

**Table 3 ijms-24-16628-t003:** Summary of the effects of acute MPD administration on experimental day 1 (ED1) to 0.6, 2.5, and 10.0 mg/kg on ventral tegmental area (VTA), locus coeruleus (LC), and dorsal raphe (DR) neuronal activity, respectively, compared to the activity of saline on ED1 BL (ED1 MPD/ED1 BL) from the all animals group; N = number of neurons in each group; ↑ column includes the number of neurons in each brain region that exhibited significant (*p* < 0.05) increases in activity after acute exposure; ↓ column includes the number of neurons in each brain region that exhibited significant (*p* < 0.05) decreases in activity after acute MPD exposure; **↔** column presents the percentage of neurons that exhibited no significant changes after acute MPD exposure. Part A contains the neuronal activity responses to the respective acute MPD doses of the three respective brain regions from all rats. Part B contains the neuronal responses to MPD from animals exhibiting behavioral sensitization. Part C contains the neuronal responses to MPD from animals exhibiting behavioral tolerance.

	ED10 MPD/ED1 MPD (Chronic)											
	VTA				LC				DR			
MPD Dose mg/kg	N	↑	↓	↔	N	↑	↓	↔	N	↑	↓	↔
** *A. Units recorded from all animals* **												
saline	45	0	2	43	56	1	3	52	57	1	2	54
0.6	141	21	19	70	134	29	54	53	137	36	35	66
2.5	142	44	34	64	146	57	35	54	142	43	46	53
10	152	93	12	47	140	110	18	12	168	78	56	34
** *B. Units recorded from sensitized animals* **												
0.6	141	24	26	1	134	15	25	94	137	34	22	81
2.5	142	62	33	47	146	63	42	41	142	36	25	81
10	152	88	40	24	140	89	40	11	168	67	59	42
** *C. Units recorded from tolerant animals* **												
0.6	141	41	20	8	134	62	33	39	137	35	31	71
2.5	142	69	44	29	146	46	30	70	142	59	42	41
10	152	61	42	49	140	91	35	14	168	66	58	44

## Data Availability

Data are contained in the medical school server.

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
