# Peer review of "Dopamine, Norepinephrine and Serotonin Participate Differently in Methylphenidate Action in Concomitant Behavioral and Ventral Tegmental Area, Locus Coeruleus and Dorsal Raphe Neuronal Study in Young Rats"

_ijms, 2023, doi:10.3390/ijms242316628_

Round 1

Reviewer 1 Report

Comments and Suggestions for Authors

The logical build-up of the manuscript is coherent, the experimental steps build on each other. It contains minimal spelling error.

In my opinion, the title of the article is too complicated. It would be useful to simplify the manuscript title.

Abbreviations should be defined when first used. It should be desirable to check out the abbreviations and their definitions once more. The definition of dorsal raphe is present in row 67, but the abbreviations is present in row 49, without definition.

Should be interesting to investigate some molecular test such as
immunohistochemisty, Western blot or PCR.

In rows 538, there is a sentence: "Twenty-nine % of VTA neurons, 64 % of LC neurons, ...". It would be useful to ues 29% form instead twenty-nine. 

The references 38, 54, and 69 are missing from the reference list. 

Author Response

Reviewer #1 suggest: C. 1: Shorten the paper title. R. 1: As suggested it was shorten. C. 2: Define abbreviation. R. 2: We define as suggested. E.g. DR definition was presented first inline 14 and repeated several times. C. 3: Suggest to add some molecular test. R. 3: The works presented in this manuscript is a summery of 5 year experimentation. We agree with the reviewer suggestion that additional molecular test will be benefit. C. 4: Change the word Twenty nine to 29 in line 538. R. 4: Since 29 is the first word in the sentence , it is common to spell the number. C. 5: References 38. 54 and 69 are missing. R. 5: The original submission did not number the references. The printers add the numbers. We aske them to take these number out.

Reviewer 2 Report

Comments and Suggestions for Authors

This manuscript by Reyes-Vasquez et al. studied the response of different brain areas after MPD treatment. I have several major concerns for this version of manuscript, which must be addressed before any acceptance to this journal.

1. The title of the paper is too long. Authors should put effort to make it more concise and accurate.

2. Is there any sex-specific differences that can be concluded from this study? Since the authors have used sufficient number of rats for this study, they should separate their analysis based on sex-differences in one of the figures.

3. In general, the figure captions are not well explained. It only explained the brief details of each figure. The ‘take home message’ of figures are not sufficiently explained.

4. In Figure 4, what are the other symbols mean, except the Asterix? This is not explained.

5. Figure 5 needs further and better explanation. Also, why there are no error bars shown in this figure?

6. The overall quality of the representation of all the figures are not very satisfactory. It seems like the authors have directly pasted these from the MS excel. Please make these figure panels better and explain well.

7. The Y-axes of all the figure panels require more descriptive, comprehensive and scientific annotations/explanations.

Author Response

Reviewer #2 suggest: C. 1: Shorten the paper title. R. 1: As suggested it was shorten. C. 2: Suggest to separate the data analysis base on sex different. R. 2: We stated in the method – line 104- that the study use male . Therefore we cannot analyze data ( female) that we did not have as suggested. C. 3: Improve the figure caption by have home message. R. 3: As suggested we add – take home message. C. 4: Some symbols are missing in Fig. 4. R. Thanks for the remark. It was added. C. 5: Fig 5 need additional detail. R. 5: As suggested, additional explanation was added. C. 6: Improve the quality of the figures. R. 6: We done the best to improve the figures. C. 7: Describe better the Y axes of the figures. R. 7: In the figures we put abbreviation in the Y axes due to space limits. In the caption we provide the detail.

Reviewer 3 Report

Comments and Suggestions for Authors

The present research article by Reyes-Vasquez et al. "Dopamine, norepinephrine and serotonin participate differently in methylphenidate (Ritalin) action. Concomitant behavioral and ventral tegmental area (VTA) locus coeruleus (LC) and dorsal raphe (DR) neuronal study in young rats" demonstrates the effect of chronic administration of different doses of MPD on animal behaviors by selectively targeting excitatory/inhibitory neurocircuits of VTA, LC and DR. Data from this study showed how each brain area responds differently to each MPD dose used, suggesting that DA, NE and 5-HT in the VTA, LC and DR exert different effects. There are no specific comments, however just curious to know why authors did not use a standard drug to counteract the behavioral effect of MPD? This would have been very informative to figure specificity of MPD in brain regions of interest in this study.

Author Response

Reviewer #3 suggest: C. 1: Why the authors did not use standard drug to counter the behavioral effects of MPD. R. 2: We use time control and saline as control since there is no drug that count out the MPD effects.

Round 2

Reviewer 2 Report

Comments and Suggestions for Authors

1. I do not see a significant edit/change in the title of the paper. The abbreviated words are eliminated only. That is not what I have asked for. Please concise the article title into one sentence. 

2. I do not understand what are the changes that the authors made in this revised version to clarify the overall message ('take home message') of each figure. Please highlight the changes. Currently there are some new lines added, which of course do not address my query/suggestion. 

3. In figure 4, it is still not well explained. The symbols other than the asterix are not explained in the figure legend. Please provide a careful edit.

4. This rebuttal still did not answer my initial query about Figure 5. Why there are no error bars shown? If you have clarified in the text, please highlight.

5. I do not see a space limit in the Y-axes and believe that proper axes can be provided well within this space. 

Author Response

1. Please concise the article title into one sentence. To our opinion the title of a paper need to tell to the reader what the paper is about. The experiment described in the paper was a concomitant behavioral and neuronal recording from three brain area recorded simultaneously following acute and chronic dose response of methylphenidate in young male rat. The title provide the reader what is the paper about. We combined the two sentence in one as suggested and took the abbreviation of the brain nuclei out. If the reviewer want to put the abbreviation back and to take the nuclei name out to make the title shorter it is o.k. with us. 2. I do not understand what are the changes that the authors made in this revised version to clarify the overall message ('take home message') of each figure. Please highlight the changes. Currently there are some new lines added, which of course do not address my query/suggestion. In our previous submission I made the changes in my computer and it did not allowed me to color it or to underline it therefore I wrote what I change in my response - Sorry my computer did not allowed me to do it. I send the paper to Dr Sack to improve the English and her remark/ improvement are in read. 3. In figure 4, it is still not well explained. The symbols other than the asterix are not explained in the figure legend. Please provide a careful edit. We respond to the reviewer, and add explanation in the last six lone of the figure legend . My computer did not aloud me to color it or to underline - sorry. 4. This rebuttal still did not answer my initial query about Figure 5. Why there are no error bars shown? If you have clarified in the text, please highlight. Figure 5, 6 and 7 provide the total number of neurons responding by significant increase or decrease . That were the % of the neurons responding significantly. All the neurons that respond significantly by increase were summed as well all the neurons that response by decrease firing rate are summed to one value, there is no SE or SD , that is the data and they have no SE or SD.

Round 3

Reviewer 2 Report

Comments and Suggestions for Authors

I am overall happy with the responses made by the authors. 

However, the authors mentioned in the latest rebuttal letter "We combined the two sentence in one as suggested and took the abbreviation of the brain nuclei out."

Am I not seeing the correct version of the revised manuscript? Where is that 'one sentence' which is a combination of two sentences? Anyway, as a reviewer I felt that the title should be concise, yet explains the whole paper.

It is upon the authors how they want to present it. I have no further comments. 

Overall, the article is a product of valid scientific experiments and logical conclusions.

Author Response

Dopamine, norepinephrine and serotonin participate differently in methylphenidate action in concomitant behavioral and ventral tegmental area, locus coeruleus and dorsal raphe neuronal study in young rats